# No Waning of Pneumococcal Vaccine Responses over Time in People with Inflammatory Arthritis: Findings from a Single Centre Cohort

**DOI:** 10.3390/vaccines12010069

**Published:** 2024-01-10

**Authors:** Deepak Nagra, Katie Bechman, Mark D. Russell, Zijing Yang, Maryam Adas, Sujith Subesinghe, Andrew Rutherford, Edward Alveyn, Samir Patel, Chris Wincup, Arti Mahto, Christopher Baldwin, Ioasaf Karafotias, Andrew Cope, Sam Norton, James Galloway

**Affiliations:** 1Centre for Rheumatic Disease, King’s College London, London WC2R 2LS, UKssubesinghe@nhs.net (S.S.); james.galloway@kcl.ac.uk (J.G.); 2King’s College Hospital NHS Trust, London SE5 9RS, UK

**Keywords:** pneumococcal vaccine, PPSV23, immunosuppression, arthritis

## Abstract

Background: Vaccination against pneumococcus reduces the risk of infective events, hospitalisation, and death in individual with inflammatory arthritis, particularly in those on immunomodulating therapy who are at risk of worse outcomes from pneumococcal disease. The objective of this study was to investigate the serological protection following vaccination against pneumococcal serovars over time. **Methods:** This was a single centre, retrospective cohort study of individuals with rheumatoid arthritis, psoriatic arthritis, or axial spondylarthritis who had previously received the PPSV23 polysaccharide pneumococcal vaccine (Pneumovax). Data were retrieved between January 2021 to August 2023. Dates of previous pneumococcal vaccination were identified using linked primary care records. Serum serotype levels were collected. The primary outcome was serological response defined as a titre ≥0.35 mcg/mL in at least five from a total of 12 evaluated pneumococcal serovars, examined using a Luminex platform. Multivariate logistic regression models adjusting for age, gender, ethnicity, co-morbidities, and the use of prednisolone, conventional synthetic and biological DMARDs were used to determine the odds of a sustained serological response according to time categorised into ≤5 years, 5–10 years, and ≥10 years since vaccination. **Results:** Serological response was measured in 296 individuals with inflammatory arthritis, with rheumatoid arthritis the most common diagnosis (74% of patients). The median time between pneumococcal vaccine administration and serological assessment was 6 years (interquartile range 2.4 to 9.9). A positive serological response to at least 5 serovars was present in 195/296 (66%) of patients. Time since vaccination did not significantly associate with serological protection compared with those vaccinated <5 years, the adjusted ORs of vaccine response was 1.15 (95% CI 0.64 to 2.07) in those 5–10 years and 1.26 (95% CI: 0.64 to 2.48) in those vaccinated over 10 years ago. No individual variable from the multivariate model reached statistical significance as an independent predictor of vaccine response, although steroid use at the time of vaccine had a consistent detrimental impact on serological immunity. **Conclusions:** We demonstrated that antibody titres following vaccination against pneumococcal serovars do not appear to wane over time. It appears more critical to focus on maximising the initial vaccine response, which is known to be diminished in this patient population.

## 1. Background

Pneumococcal disease is a significant public health concern due to its diverse clinical manifestations, which include pneumonia, invasive pneumococcal disease (IPD), septic arthritis, and meningitis. It is caused by Streptococcus pneumoniae, a Gram-positive bacterium, with transmission occurs via respiratory droplets. Over 90 serovars of this bacterium have been identified, each contributing differently to the disease burden. Invasive pneumococcal disease (IPD) is defined by the presence of Streptococcus pneumoniae in sites of sterility such as blood, pleural, cerebrospinal, and synovial fluid.

Pneumonia is one of the major causes of mortality in patients with Inflammatory Arthritis (IA [1]). Individual with chronic inflammatory diseases, particularly those on immunomodulating therapy are at an increased risk of pneumococcal infection [2] and have worse outcomes from the disease [3,4]. The risk of infection and IPD can be reduced through vaccination against pneumococcus. This has been demonstrated in individuals with inflammatory arthritis with significant reductions in the frequency of infective events, hospitalisation, and death in vaccinated cohorts [5]. In the United Kingdom, it is estimated that over 43,000 hospital admissions have been avoided through vaccination efforts [6].

Pneumococcal polysaccharide vaccines (PPSVs), such as the 23-valent PPSV23, work by inducing an immune response against the polysaccharide capsule of Streptococcus pneumoniae. These capsules are key virulence factors, enabling the bacteria to evade phagocytosis. The vaccine contains purified capsular polysaccharides from various serotypes, which, when introduced into the body, stimulate B cells directly without the assistance of T cells, leading to a T-cell independent immune response. This response results in the production of serotype-specific antibodies, primarily IgM and some IgG, which enhance opsonisation and phagocytosis of the bacteria. However, because this process does not involve T cells, it does not generate a strong memory B cell response, which is a crucial aspect of long-lasting immunity. This mechanism underpins the need for periodic booster vaccinations to sustain protective antibody levels, especially in populations with altered immune responses, such as patients with inflammatory arthritis [7,8].

Current guidelines advocate for the vaccination of all patients prescribed immunosuppression for rheumatic diseases, ideally before commencing therapy. Two different vaccine types are available; the polysaccharide PPSV23 vaccine and the conjugated PCV7, PCV10, PCV13, PCV15, or PCV20 vaccines [9]. There are variances in international recommendation with some countries endorsing a single dose, while others support a five-year booster. Despite the vaccine’s availability, uptake is suboptimal, particularly among rheumatic disease patients under 65 years of age who do not qualify for vaccination based on age criteria alone [10]. A recent systematic review of pneumococcal vaccine responses in the context of immune suppression demonstrated that multiple treatments used for inflammatory arthritis blunt vaccine responses [11].

The COVID-19 pandemic catalysed a surge in vaccine research, leading to a remarkable expansion in our understanding of vaccine-induced immune responses. A key observation has been the correlation between serological markers of vaccine response and the subsequent risk of infection. Enhanced methodologies and large-scale studies have provided much clearer evidence that the presence and magnitude of specific antibodies serve as reliable predictors of immunity against COVID-19 infection. Whilst this has always been considered plausible, robust evidence was lacking.

In parallel, knowledge about the waning of immunity post-vaccination has advanced. Studies have consistently shown that while COVID-19 vaccines elicit a robust initial immune response, this response tends to diminish over time [12]. Of particular interest is the observation that this waning of immunity is more pronounced in individuals on immunosuppressive therapy. The COVID-19 pandemic has provided a wealth of data that illustrate the varying vaccine response in immunocompromised individuals, including those with autoimmune diseases, organ transplant recipients, and patients undergoing cancer treatment.

Vaccination against Streptococcus pneumoniae induces a robust initial immune response characterised by the production of capsule-specific antibodies [13]. These antibodies play a crucial role in targeting the polysaccharide capsule of the pneumococcus, which is essential for its pathogenicity. Typically, antibody titres reach their peak between a few weeks and a few months following vaccination, offering substantial protection against pneumococcal infections [14]. Over time, a decline in these antibody titres is expected. Waning of immunity can be attributed to the natural life cycle of plasma cells, which are the primary producers of antibodies. Once the immediate threat of infection is neutralized, the body downregulates the production of specific antibodies to conserve resources for future immune challenges. Additionally, the lack of ongoing antigenic stimulation, as would occur in natural infections, results in a gradual decrease in the number of circulating memory B cells specific to the pneumococcal capsule. Consequently, the diminished presence of these memory cells and the reduced antibody levels necessitate periodic booster vaccinations to maintain protective immunity against pneumococcal disease. However, information on the trajectory of the immune response following pneumococcal vaccination is scarce, especially amongst immunocompromised populations.

While COVID-19 vaccine research has greatly advanced our understanding of vaccine-induced immunity, particularly in the context of immunosuppression, such comprehensive insights are lacking for pneumococcal vaccines. Given the increased risk and severity of pneumococcal disease in individuals with inflammatory arthritis, especially those on immunomodulatory therapies, our study sets out to address part of this knowledge gap [15,16].

This study aimed to investigate the hypothesis that serological protection against pneumococcal serovars diminishes over time in this patient group [17] and to examine factors associated with vaccine response.

## 2. Methods

### 2.1. Study Design and Population

We conducted a retrospective cohort study at King’s College Hospital, evaluating adults with inflammatory arthritis (RA, PsA, AxSpA). Adults with an inflammatory arthritis (Rheumatoid Arthritis [RA], Psoriatic Arthritis [PsA], and Axial Spondylarthritis [AxSpA]) who had previously received the 21-valent polysaccharide pneumococcal vaccine PPSV23 (Pneumovax) were followed up from January 2021 to August 2023

### 2.2. Primary Outcome

The primary outcome assessed was serological response to pneumococcal vaccine. This was defined as a titre ≥0.35 mcg/mL in more than five of a total of 12 evaluated pneumococcal serovars (1, 4,5, 6b, 9v, 14, 18c, 19f, 23f, 3, 7f, 19a) as defined by the WHO [18,19]. Pneumococcal serovars were examined using the Luminex laboratory platform.

Dates of previous pneumococcal vaccination were identified using linked primary care records, with additional information regarding demographics and possible confounders (conventional synthetic disease-modifying antirheumatic drugs (csDMARDs), biologics DMARD’s (bDMARD) and targeted DMARD’s (tDMARD), steroid use, disease duration, and comorbidities obtained from both primary and secondary care records) immediately before vaccination. csDMARD’s include methotrexate, leflunomide, sulfasalazine and hydroxychloroquine, whilst bDMARD’s include Tumour necrosis factor inhibitors (TNFi [adalimumab, certolizumab-pegol, etanercept, golimumab and infliximab.]), Interlukin-6 inhibitors (IL6i [tocilizumab and Sarilumab]), Cytotoxic T-lymphocyte antigen 4 inhibitors (CTLA-4i [abatacept]), and Rituximab and tDMARD include Janus Kinase Inhibitors (JAKi [Baricitinib, Filgotinib, Tofacitinib and Upadacitinib]).

### 2.3. Statistical Methodology

The target sample size was 400 patients, based on this providing 90% power to detect an odds ratio of at least two for a binary variable using logistic regression, assuming a 30% response rate at the 5% significance level. The actual sample size achieved was lower than anticipated. With 296 patients, power is reduced from 90% to 81%, which we deemed to be sufficient to progress with the analysis as planned.

Vaccine responses were graphically examined by plotting log antibody titre against time as a continuous variable. Logistic regression models were used to determine the odds of a sustained serological response according to time, categorised into ≤5 years, 5–10 years, and ≥10 years since vaccination. This was adjusted for potential confounders decided a priori including age, gender, ethnicity, lung disease, renal disease, cancer, diabetes, and csDMARD and t/bDMARD use. Sensitivity analyses explored responses for individual pneumococcal serovars.

We minimised the levels of missing information through meticulous data extraction from our single centre cohort. As a result, no modelling for missing data was necessary; instead, we introduced an “unknown” variable code to retain individuals in the regression model [20].

### 2.4. Ethics

Ethical approval was granted under the local review board’s approval for a quality improvement project aimed at identifying and addressing gaps in pneumococcal vaccination. The secondary use of anonymised data for this study were in line with the Health Research Authority (HRA) decision tool and National Research Ethics Service (NRES) guidance, which confirmed that national ethics approval was not requisite for our project.

## 3. Results

### 3.1. Characteristics of Cohort

Pneumococcal serological response was measured in 296 individuals with inflammatory arthritis with RA being the most common diagnosis (74% vs. 13% with AxSpA and 13% with PsA) (Table 1). The mean age was 55 years, and the cohort were predominantly female (75%), consistent with the epidemiology of inflammatory conditions. There was a high proportion of non-white ethnicities, which is representative of our local population demography. Asians accounted for 13%, and Black participants made up 17% of the total cohort.

### 3.2. Time between Pneumococcal Vaccine Administration and Serological Assessment

The median time between pneumococcal vaccine administration and serological assessment was 6 years (interquartile range 2.4 to 9.9). Only 9 individuals had received a booster vaccine. Forty three percent of individuals had received a pneumococcal vaccination in the last 5 years, 33% were vaccinated 5 to 10 years ago, and 24% were vaccinated ≥10 years ago. Individuals vaccinated less than five years ago were younger than those vaccinated over ten years ago (51 vs. 62 years old). The comorbidity burden was also lower in this group [RDCI score ≤5 years: 1.7 vs. ≥10 years 2.4], with a lower proportion of patients with hypertension diabetes and ischaemic heart disease compared with those vaccination over ten years ago [hypertension: 25% versus 51%, diabetes 13% versus 29% and ischaemic heart disease 11% vs. 21%]. Prior cancer was reported in 7% of the cohort, with no significant fluctuation across the different groups. There was a marked decrease in the use of DMARDs at the time of vaccination in those vaccinated more than 10 years ago [21% compared with 49% on DMARDs at the time of vaccination in the <5 year since vaccine group]. Biologic treatments were most common in the <5 years group.

### 3.3. Serological Response

Serological response to at least 5 serovars was present in 195/296 (66%) of patients. The likelihood of a robust serological response to the pneumococcal vaccine did not significantly associated with time since vaccination in both the age and sex adjusted model and the fully adjusted model. Compared with those vaccination within 5 years, the adjusted odds ratio for a vaccine response was 1.15 (95% CI 0.64 to 2.07) in those vaccinated between 5 and 10 years and OR 1.26 (95% CI: 0.64 to 2.48) in those vaccinated over 10 years ago (Figure 1).

We did not identify any other individual variable from the multivariate model that reached statistical significance as an independent predictor of vaccine response. Patients who had not been treated with DMARDs at the time of vaccination (DMARD-naive) showed a higher point estimate for the odds of a positive serological response (OR: 1.33, 95% CI: 0.78 to 2.26). Conversely, patients who had not received biologic therapies (biologic-naive) exhibited a lower point estimate for the odds of a robust serological response (OR: 0.60, 95% CI: 0.30 to 1.19). The effect of individual biologics was not examined due to the small numbers in each group. The use of prednisolone also associated with a lower point estimate for the odds of a robust serological response (OR: 0.36, 95% CI: 0.12 to 1.05), although again this did not reach statistical significance.

Individual pneumococcal serovar vaccine responses were examined against time using scatter plots with a line of best fit derived from a linear regression (Figure 2). There was no convincing trend of decline in the titre of serovar response over time since vaccination, contrary to our original hypothesis that vaccine’s efficacy may wane over time. While outliers are present, with some individuals exhibit exceptionally high or low antibody titre, these do not detract from the overall pattern indicating sustained immune response. This observation holds true across the different serovars shown, although the level of response to each serovar is not uniform.

### 3.4. Sensitivity Analysis

We conducted sensitivity analyses exploring predictors of individual serovar responses. These showed similar patterns for all serovars, with no evidence of time since vaccination associating with serological response. Looking across the individual serovars, the negative association of prednisolone use at the time of the vaccine was consistent, with 3 of the 12 serovars having significant *p* values < 0.05 and substantial effect sizes (OR ≤ 0.3). 

We also performed sensitivity analyses limiting the cohort to those without booster pneumococcal vaccine (n = 287, 97%). This demonstrated similar results to the primary analysis, although the negative association with prednisolone and vaccine response reached statistical significance [adjOR 0.30 (95% CI: 0.10 to 0.93).

## 4. Discussion

Our study set out to examine the durability of pneumococcal vaccine responses in a cohort of patients with inflammatory arthritis. Contrary to our initial hypothesis that the effectiveness of the vaccine would wane over time [21,22,23], we found no evidence to support this. In fact, the serological response observed did not associate with time since vaccination. We observed a pneumococcal vaccine response in two thirds of our cohort, which aligns with expectations for an inflammatory arthritis population and suggests our cohort is representative. These findings challenge the notion that pneumococcal vaccine efficacy decreases over time. Despite adjustments for multiple confounders, these findings remained robust, reinforcing the validity of our results.

To date, studies have shown that antibody titres wane with time in both adults and children [24]. In contrast to our study, two large randomised controlled trials in the paediatric population demonstrated certain serotypes have shown to wane faster than others including serotypes 4, 9V, 19F and 23F (NRT2316, NRT3069 [25]). A previous study (N = 214) comparing serotype concentrations in both immunocompetent and immunosuppressed individuals suggested waning of antibody titres in both groups with titres beginning to drop by 19% at 12 months following vaccination [26].

Our findings contribute to the broader understanding of vaccine responses in immunocompromised populations, a topic that has gained significant attention during the COVID-19 pandemic. Similar to observations with COVID-19 vaccines, our study underscores the variability of vaccine-induced immunity in individuals on immunomodulatory therapies [27]. Research on COVID-19 vaccines has shown that while immunocompromised patients can mount an immune response, the magnitude and durability of this response are often reduced compared with healthy individuals [28]. This parallels our observations with pneumococcal vaccines, where the primary challenge appears to be achieving a robust initial response. The insights gained from COVID-19 vaccine studies, particularly regarding strategies to enhance initial vaccine responses, such as dose adjustments and additional booster doses, could be valuable in optimising pneumococcal vaccination strategies for patients with inflammatory arthritis.

The phenomenon of waning immunity over time, as observed in COVID-19 vaccines, offers a pertinent parallel to our findings on pneumococcal vaccine responses in patients with inflammatory arthritis. Studies have demonstrated a significant decrease in humoral responses six months post-COVID-19 vaccination, particularly among specific subgroups like older individuals and those with immunosuppression. This contrasts with our observations where pneumococcal vaccine responses remained relatively stable over time. The comparative analysis underscores the variability in immune response dynamics across different types of vaccines. While COVID-19 vaccines show a rapid decline in antibody levels necessitating booster doses, our study suggests that pneumococcal vaccines may elicit a more durable response in the inflammatory arthritis population. This divergence in immune response longevity between different vaccines emphasizes the need for disease-specific vaccination strategies and potentially different booster policies. Understanding these differences is crucial for optimizing vaccine efficacy in immunocompromised patients, as the waning efficacy observed in COVID-19 vaccines may not directly translate to pneumococcal vaccines, highlighting the uniqueness of each vaccine’s interaction with the immune system [29]. When considering other subunit vaccines and those more similar in structure to the pneumococcal vaccine, such as Hepatitis B vaccine, there is little published on waning immune response in adult populations. The efficacy of the Hepatitis B vaccine has been examined in those with rheumatoid arthritis with demonstration that both age and B cell depleting therapies impaired vaccine response [30].

When considering the reasons for the heterogenicity of results compared with this study, an important factor has been the definition of a serotype response. The above studies defined serotype response of over 1.3 µg/mL in 70% of more of the available serotypes [26]. This metric differs from the recommendations of WHO and is considered the outcome measure of choice in patient’s being tested for a pneumococcal vaccine response with immunodeficiency syndromes despite limited evidence to support its validation [31,32]. Seroprotection has been defined as and validated as an antibody titre of over 0.35 ug/mL on the Luminex platform [33], despite this differences in the use and clinical interpretation of the pneumococcal antibodies remain [34]. The WHO consensus defines antibody response as being >0.35 µg/mL and this is further supported by the European Medicines Agency (EMA) [35]. Discrepancies remain on the exact definition of a vaccine response however a commonly used metric in studies including patients with rheumatic disease is a response to six serotypes [36].

In exploring explanations for why pneumococcal vaccine responses do not wane over time, we considered the understanding that immune responses mature over time. Unlike the rapid waning in immunity to with COVID-19 vaccines [37], immune activation with polysaccharides vaccines, such as the pneumococcal, are stronger with more sustained B cell and antibody responses [8]. It is possible that our binary approach to defining vaccine response, using a cut-off titre of 0.35, might have obscured subtle declines in vaccine-induced immunity.

In this study, we adhered to the UK standardised definition of seroprotection following pneumococcal vaccination. It is important to note, however, that internationally, definitions of seroprotection can vary significantly. This variation underscores the challenges in achieving a consensus on protective antibody levels, which are often extrapolated from epidemiological data. A limitation of our approach is the lack of data on functional antibody activity, which is particularly relevant in our population. The functionality of antibodies, especially in terms of their opsonophagocytic activity, is crucial for effective immunity against pneumococcal infection. This gap in data might limit our understanding of the true protective capacity of the observed antibody titres, especially in individuals with altered immune responses. Our approach also applied a uniform standard for seroprotection across all pneumococcal serotypes. This approach, while methodologically convenient, may oversimplify the immunological landscape of pneumococcal disease. In reality, different serotypes may require varying antibody titres to confer protection, a nuance that our current definition of seroprotection does not capture. This oversimplification could potentially obscure serotype-specific vulnerabilities. However, we attempted to explore this by plotting serovar responses individually and our data did not reveal any clear pattern of reduced response over time specific to any individual serovar.

We observed trends suggesting that certain patient factors, such as the use of prednisolone, may influence variation in vaccine responses. Given that this was not the primary focus of our study, these observations are considered exploratory and require robust replication. Nonetheless, the association with prednisolone use is in line with existing knowledge about its immunosuppressive effects [38].

Our study’s strengths lie in its adequately powered sample size for our primary question, as well as the robustness of the serological assessment and consistency of definitions with the current literature. We acknowledge limitations in our study, including the inability to scrutinise individual drug exposures due to small subgroup sizes, particularly concerning drugs like rituximab known for their profound impact on vaccine responsiveness. Additionally, although the categorisation of patients into different time intervals since vaccination allows the data to be shared in a manner that is clinically interpretable, using these time windows may have reduced our statistical power. However, this would not plausibly account for the lack of observed waning in vaccine titres.

Our study challenges the traditional approach of using binary cut points for vaccine response, particularly the low thresholds currently accepted as indicative of seroprotection. In light of our findings and lessons from COVID-19, where higher antibody titres correlate with better protection, it becomes evident that aiming for a strong primary response is crucial. The current UK standard thresholds, while widely used, may be insufficient for optimal protection, especially in immunocompromised populations. This suggests a need to revisit our vaccination strategies, focusing on enhancing the initial response rather than solely relying on periodic boosters. Strategies could include dose adjustments, alternative vaccine formulations, or additional booster doses, tailored to the unique immunological challenges faced by patients with autoimmune diseases.

A limitation to consider in this cohort is the possibility of reinfection following vaccine administration which could interfere with antibody titres, a variable we are unable to control or adjust for.

Our findings raise questions for future research. One area of interest is the potential benefit of a prime-boost vaccine regimen combining conjugate and polysaccharide vaccines in patients with inflammatory arthritis. Conjugate vaccines, which link polysaccharides to a protein carrier, elicit a T-cell dependent immune response, potentially leading to a more robust and long-lasting immunity, including the generation of memory B cells. This approach could address the limitations of polysaccharide vaccines, particularly in achieving a stronger initial immune response. Future studies should explore the efficacy, safety, and optimal scheduling of such combined regimens, potentially transforming the approach to pneumococcal vaccination in immunocompromised populations.

## 5. Conclusions

In conclusion, our findings suggest that the hypothesis of waning vaccine titres over time is incorrect. It appears more critical to focus on maximising the initial vaccine response, which is known to be diminished in this patient population. The value of repeated vaccination over time, based on our data, may be limited. Future efforts should be directed towards strategies that enhance the primary response to vaccination in patients with autoimmune diseases.

## Figures and Tables

**Figure 1 vaccines-12-00069-f001:**
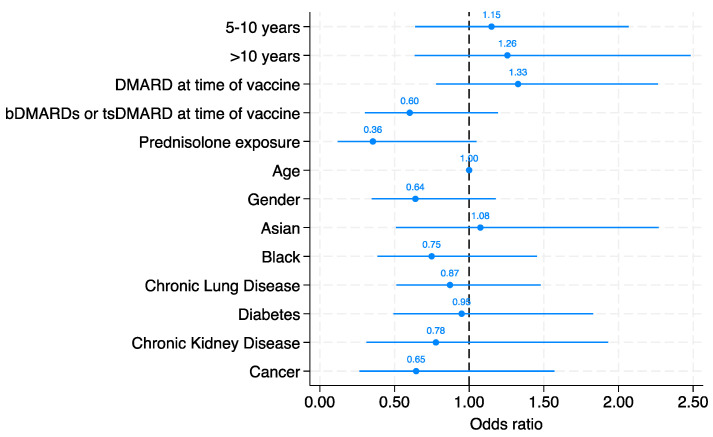
Multivariate logistic regression model demonstrating the likelihood of vaccine response based on time since vaccine, DMARD, bDMARD, and tsDMARD use at time of vaccine, prednisolone exposure, age, gender, ethnicity, and co-morbidities. The odds ratio (OR) for each parameter is illustrated in the figure.

**Figure 2 vaccines-12-00069-f002:**
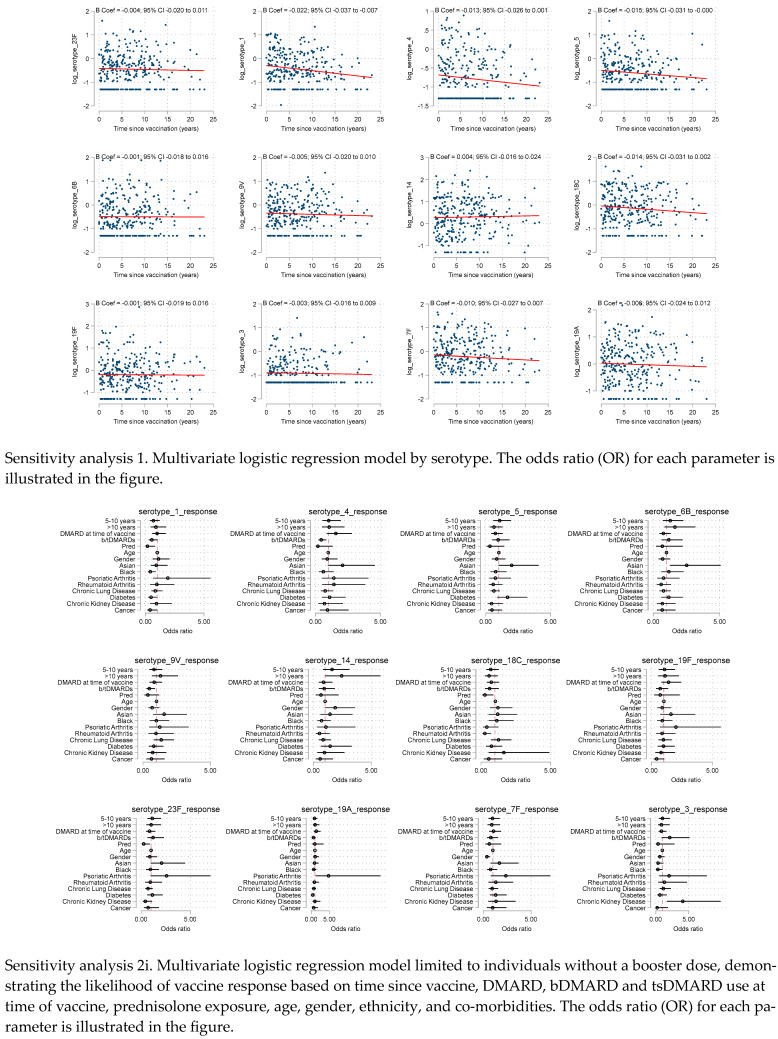
Scatter plot for each individual serotype demonstrating the log transform serotype response against time in years, with a line of best fit derived from a linear regression plot.

**Table 1 vaccines-12-00069-t001:** Baseline characteristics.

	Time Since Vaccination			
	<5 Years	5–10 Years	>10 Years	Total
	N = 127	N = 97	N = 72	N = 296
Baseline Characteristics				
Age	51.0 (14.9)	56.1 (14.8)	61.7 (15.9)	55.3 (15.6)
Female	87 (69.0%)	77 (79.4%)	55 (77.5%)	219 (74.5%)
Ethnic Group				
Asian	21 (16.5%)	10 (10.3%)	8 (11.1%)	39 (13.2%)
Black	24 (18.9%)	16 (16.5%)	11 (15.3%)	51 (17.2%)
White	82 (64.6%)	71 (73.2%)	53 (73.6%)	206 (69.6%)
Rheumatic Diseases Comorbidity Index	1.7 (1.8)	2.0 (2.0)	2.4 (1.8)	2.0 (1.9)
Chronic Lung Disease	41 (32.3%)	37 (38.1%)	26 (36.1%)	104 (35.1%)
Heart Attack	18 (14.2%)	12 (12.4%)	17 (23.6%)	47 (15.9%)
Ischaemic Heart Disease	15 (11.8%)	8 (8.2%)	15 (20.8%)	38 (12.8%)
Stroke	2 (1.6%)	1 (1.0%)	0 (0.0%)	3 (1.0%)
Hypertension	32 (25.2%)	32 (33.0%)	37 (51.4%)	101 (34.1%)
Previous Fracture	9 (7.1%)	4 (4.1%)	5 (6.9%)	18 (6.1%)
Depression	27 (21.4%)	18 (18.6%)	20 (27.8%)	65 (22.0%)
Diabetes	17 (13.4%)	16 (16.5%)	21 (29.2%)	54 (18.2%)
Cancer	6 (4.7%)	9 (9.3%)	7 (9.7%)	22 (7.4%)
Peptic Ulcer Disease	25 (19.7%)	23 (24.0%)	13 (18.1%)	61 (20.7%)
Chronic Kidney Disease	6 (4.7%)	10 (10.3%)	9 (12.5%)	25 (8.4%)
Diagnosis				
Axial Spondylarthritis	20 (15.7%)	11 (11.3%)	7 (9.7%)	38 (12.8%)
Psoriatic Arthritis	20 (15.7%)	12 (12.4%)	6 (8.3%)	38 (12.8%)
Rheumatoid Arthritis	87 (68.5%)	74 (76.3%)	59 (81.9%)	220 (74.3%)
Rheumatoid arthritis patients only				
Seronegative	20 (23.0%)	25 (33.8%)	18 (30.5%)	63 (28.6%)
RF only	11 (12.6%)	8 (10.8%)	11 (18.6%)	30 (13.6%)
CCP only	10 (11.5%)	7 (9.5%)	4 (6.8%)	21 (9.5%)
Double (RF & CCP) antibody positive	46 (52.9%)	34 (45.9%)	26 (44.1%)	106 (48.2%)
Treatment at time of vaccine				
Prednisolone at time of vaccine	10 (7.9%)	5 (5.2%)	0 (0.0%)	15 (5.1%)
DMARD at time of vaccine	62 (48.8%)	49 (50.5%)	15 (20.8%)	126 (42.6%)
Biologic at time of vaccine	36 (28.3%)	16 (16.5%)	6 (8.3%)	58 (19.6%)
- Biologic monotherapy	21 (16.5%)	8 (8.2%)	3 (4.2%)	32 (10.8%)
- Biologic + DMARDs combination therapy	15 (11.8%)	8 (8.2%)	3 (4.2%)	26 (8.8%)

## Data Availability

Data for this project belong to the Department of Rheumatology, King’s college Hospital and the Centre for Rheumatic Disease, King’s College London.

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
