# Peer review of "No Waning of Pneumococcal Vaccine Responses over Time in People with Inflammatory Arthritis: Findings from a Single Centre Cohort"

_vaccines, 2024, doi:10.3390/vaccines12010069_

Round 1
Reviewer 1 Report
Comments and Suggestions for Authors
Dear Authors!
Thank you for the opprotunity to review the manuscript
The manuscript is actual. Pneumococcus infection is a serious complication of immune mediated disease and vaccination of such patients should be encourage between healthcare providers and patients as well as routine antibodies assessment every several years
The results and statistical anaylisis is reproducible
The Discussion contains all contemporary literature
The Conclusion emphasises the study results and corresponds with title and the study's aim
Minor suggestions
1) Part of RA 74 (in the text and in the table) and 75 in the abstract.
2) May be better use RF not RhF in table 1
3) May be better use Double (RF+CCP) antibody positive instead of Double antibody positive
4) List line of Table 1. Biologics only or with/without DMARD. May be better to split on two lines Biologics alone and Biologics+DMARDs
5) Did patients receive corticosteroids?
6) Can you add information about disease duration before vaccination and study inclusion?
7) How many patients were vaccinated before and after the onset of the rheumatic disease?
8) Were there any patients in your cohort who were vaccinated against pneumococcus in childhood?
Author Response
1) Part of RA 74 (in the text and in the table) and 75 in the abstract.
Many thanks. Percentage value corrected to 74% in the abstract.
2) May be better use RF not RhF in table 1
Many thanks, I have changed this.
3) May be better use Double (RF+CCP) antibody positive instead of Double antibody positive
Many thanks, I have changed this.
4) List line of Table 1. Biologics only or with/without DMARD. May be better to split on two lines Biologics alone and Biologics+DMARDs
Thank you for this comment. We have now included two rows at the bottom on the table with biologics alone and biologics + DMARD. We feel it is important to include the variables above this (DMARD at time of vaccine and Biologic at time of vaccine) as these were used in our models, and therefore should be presented in the table.
5) Did patients receive corticosteroids?
Yes, a small number of patients received prednisolone at the time of vaccination (n=15, 5%). We have now included this data in our baseline table. We do not have accurate date on prednisolone dose due to this being a retrospective study, and we cannot be entirely sure of the steroid dose at time of vaccination. We did examine prednisolone exposure in relation to vaccine response and found that it did not associated with vaccine response in multivariate logistic regression (figure A1). However, when looking across the individual serovars, we did find a negative association of prednisolone use at the time of the vaccine (Supplementary 1).
6) Can you add information about disease duration before vaccination and study inclusion?
Unfortunately, we do not have accurate date of diagnosis in our rheumatology database. This is because our database was only started in 2010, and for a number of patients (around 1/3rd of the cohort) a diagnosis data was given at this start date rather than their true diagnosis date. We have found errors in the diagnosis date for numerous patients and therefore chose not to use this data. We did try to extrapolate a diagnosis date using GP local care records but found issues with this as well. This was not an issue with DMARD and biologic prescribing where the dates are generated from electronic prescribing or correctly entered in GP local care records. We appreciate that data on disease duration would be helpful, but we feel this would not have a major impact on the main questions of our paper, which examines vaccine response over time.
7) How many patients were vaccinated before and after the onset of the rheumatic disease?
Unfortunately, we have the same issue as described above, and were unable to identify dates of diagnosis for 1/3 of the cohort.
8) Were there any patients in your cohort who were vaccinated against pneumococcus in childhood?
Of the pneumococcal vaccines that we have presented in this study, none were given as part of the childhood vaccination schedule. We do not have additionally data on the number of patients who had a pneumococcal vaccine as part of their childhood vaccination schedule.
Reviewer 2 Report
Comments and Suggestions for Authors
The article refers to the humoral response against neumococcal vaccine in patients with inflammatory arthritis. The article is well written and the oaim and the objectives are clear. However, there are some points to consider. The response to neuumococcal vaccine, as des ribed by the authors in a prevous publication, is a good response; it can not be compared to vaccines against Sars cov2 virus since they differ in several aspects. It could be compared though with other schemes like hepatitis B virus recombinant protein see PMID: 31764493. In that cohort age is related to vaccine response; however, the conditions differ. Another issue is that Asian patient seem to respond more to the vaccine although is not significant. Is the.MHC iand antigen presentation nvolved in such difference? See https://doi.org/10.1016/j.ijid.2019.12.021
Finally, the study should include as a limitation the lack of information of possible reinfection of the indivuals which may contribute to the antibody levels detrcted.
Comments on the Quality of English LanguageThe English language is fine.
Author Response
The article refers to the humoral response against pneumococcal vaccine in patients with inflammatory arthritis. The article is well written and the aim and the objectives are clear. However, there are some points to consider.
1) The response to pneumococcal vaccine, as described by the authors in a previous publication, is a good response; it can not be compared to vaccines against Sars cov2 virus since they differ in several aspects. It could be compared though with other schemes like hepatitis B virus recombinant protein see PMID: 31764493. In that cohort age is related to vaccine response; however, the conditions differ.
Many thanks. We acknowledge that the sars-cov-2 vaccine and the pneumococcal vaccine are not alike and differ in many aspects. Due to the recent pandemic, a great deal of work has been done to evaluate the use of covid-19 vaccines in those with rheumatic disease. As a result, the work from these studies has been discussed and referenced as a benchmark towards vaccine effectiveness in those who are immunosuppressed. We do appreciate that the two cannot be directly compared and rather we embark to discuss a broader understanding of vaccine response (outlined in line 259 onwards). We highlight each individual vaccine is unique (line 285). On your advice, we have included the suggestion offered by you on hepatitis B vaccinations and discussed vaccine response over time with this vaccination. (line 288) – many thanks.
2) Another issue is that Asian patient seem to respond more to the vaccine although is not significant. Is the.MHC iand antigen presentation involved in such difference? See https://doi.org/10.1016/j.ijid.2019.12.021
Many thanks. We acknowledge the work done by Choe Et Al. Although we do recognise the observation of Asian ethnicity and vaccine response, we feel that no firm conclusion can be obtained from our study results. Although interesting, and potential an area for future work, we did not want to mislead the reader by discussing the associations between ethnicity and vaccine response, as we did not find a strong association in our study.
3) Finally, the study should include as a limitation on the lack of information of possible reinfection of the individuals which may contribute to the antibody levels detected.
Many thanks for pointing this out. We agree this is an important consideration and have now included the addition of the following (line 354 )
‘A limitation to consider in this cohort is the possibility of reinfection following vaccine administration which could interfere with antibody titres, a variable we are unable to control or adjust for’